# Migratory Wild Birds as a Potential Disseminator of Antimicrobial-Resistant Bacteria around Al-Asfar Lake, Eastern Saudi Arabia

**DOI:** 10.3390/antibiotics10030260

**Published:** 2021-03-05

**Authors:** Ibrahim Elsohaby, Ahmed Samy, Ahmed Elmoslemany, Mohammed Alorabi, Mohamed Alkafafy, Ali Aldoweriej, Theeb Al-Marri, Ayman Elbehiry, Mahmoud Fayez

**Affiliations:** 1Department of Animal Medicine, Faculty of Veterinary Medicine, Zagazig University, Zagazig City 44511, Egypt; 2Department of Health Management, Atlantic Veterinary College, University of Prince Edward Island, Charlottetown, PE C1A 4P3, Canada; 3Reference Laboratory for Veterinary Quality Control on Poultry Production, Animal Health Research Institute, Agricultural Research Center, Dokki, Giza 12618, Egypt; dr.ahmed189@gmail.com; 4Immunogenetics, The Pirbright Institute, Surrey GU24 0NF, UK; 5Hygiene and Preventive Medicine Department, Faculty of Veterinary Medicine, Kafrelsheikh University, Kafr El-Sheikh 33516, Egypt; aelmoslemany@gmail.com; 6Department of Biotechnology, College of Science, Taif University, P.O. Box 11099, Taif 21944, Saudi Arabia; maorabi@tu.edu.sa (M.A.); m.kafafy@tu.edu.sa (M.A.); 7Veterinary Health and Monitoring, Ministry of Environment, Water and Agriculture, Riyadh 11195, Saudi Arabia; Dr.alivet@mewa.gov.sa; 8Al-Ahsa Veterinary Diagnostic Lab, Ministry of Environment, Water and Agriculture, Al-Ahsa 31982, Saudi Arabia; theep8@hotmail.com (T.A.-M.); mahmoudfayez30@hotmail.com (M.F.); 9Department of Bacteriology, Mycology and Immunology, Faculty of Veterinary Medicine, University of Sadat City, Sadat City 32897, Egypt; aymanella2007@yahoo.com; 10Department of Public Health, College of Public Health and Health Informatics, Qassim University, Al Bukayriyah 52741, Saudi Arabia; 11Department of Bacteriology, Veterinary Serum and Vaccine Research Institute, Ministry of Agriculture, Cairo 131, Egypt

**Keywords:** antimicrobial resistance, *E. coli*, *Salmonella*, *Staphylococcus*, migratory wild birds, multidrug

## Abstract

Migratory wild birds acquire antimicrobial-resistant (AMR) bacteria from contaminated habitats and then act as reservoirs and potential spreaders of resistant elements through migration. However, the role of migratory wild birds as antimicrobial disseminators in the Arabian Peninsula desert, which represents a transit point for birds migrating all over Asia, Africa, and Europe not yet clear. Therefore, the present study objective was to determine antimicrobial-resistant bacteria in samples collected from migratory wild birds around Al-Asfar Lake, located in Al-Ahsa Oasis, Eastern Saudi Arabia, with a particular focus on *Escherichia coli* virulence and resistance genes. Cloacal swabs were collected from 210 migratory wild birds represent four species around Al-Asfar. *E. coli*, *Staphylococcus*, and *Salmonella* spp. have been recovered from 90 (42.9%), 37 (17.6%), and 5 (2.4%) birds, respectively. Out of them, 19 (14.4%) were a mixed infection. All samples were subjected to AMR phenotypic characterization, and results revealed (14–41%) and (16–54%) of *E. coli* and *Staphylococcus* spp. isolates were resistant to penicillins, sulfonamides, aminoglycoside, and tetracycline antibiotics. Multidrug-resistant (MDR) *E. coli* and *Staphylococcus* spp. were identified in 13 (14.4%) and 7 (18.9%) isolates, respectively. However, none of the *Salmonella* isolates were MDR. Of the 90 *E. coli* isolates, only 9 (10%) and 5 (5.6%) isolates showed the presence of *eaeA* and *stx2* virulence-associated genes, respectively. However, both *eaeA* and *stx2* genes were identified in four (4.4%) isolates. None of the *E. coli* isolates carried the *hlyA* and *stx1* virulence-associated genes. The *E. coli* AMR associated genes *bla_CTX-M_*, *bla_TEM_*, *bla_SHV_*, *aac(3)-IV*, *qnrA*, and *tet(A)* were identified in 7 (7.8%), 5 (5.6%), 1 (1.1%), 8 (8.9%), 4 (4.4%), and 6 (6.7%) isolates, respectively. While the *mecA* gene was not detected in any of the *Staphylococcus* spp. isolates. Regarding migratory wild bird species, bacterial recovery, mixed infection, MDR, and AMR index were relatively higher in aquatic-associated species. Overall, the results showed that migratory wild birds around Al-Asfar Lake could act as a reservoir for AMR bacteria enabling them to have a potential role in maintaining, developing, and disseminating AMR bacteria. Furthermore, results highlight the importance of considering migratory wild birds when studying the ecology of AMR.

## 1. Introduction

Migratory and resident wild birds are important reservoirs and spreaders of zoonotic and antimicrobial-resistant (AMR) bacteria [1]. About 5 billion migratory wild birds fly across continents twice a year [2], facilitating the global transfer of several pathogens [3]. Different pathogenic bacterial species were isolated from wild birds, including *Escherichia coli* (*E. coli*) [4], *Salmonella* [5], *Staphylococcus* spp. [6], *Campylobacter* [7], and *Listeria monocytogenes* [8]. Indirect transmission of these pathogens to humans has also been reported [9].

AMR is a dynamic and multifaceted One Health problem involving humans, animals, and the environment [10]. Even though the exact mechanism of environmental dissemination of AMR is not fully understood, existing research revealed the central role of human factors [11]. Furthermore, several growing pieces of evidence indicate the ability of migratory wild birds to transport resistant elements to regions away from their anthropogenic origin [12].

The uncontrolled use of antimicrobial therapy in veterinary medicine and humans [13] leads to the discharge of AMR bacteria to untreated sewage, livestock farms, wastewater treatment facilities, aquaculture ponds, and landfills [14,15,16]. The discharged AMR bacteria find their way to the migratory wild bird habitats representing extra selective pressure for resistant bacteria in addition to the risk for long-distance dispersal to unexposed wildlife and free-range animal [12]. The resulting proliferation and dissemination of AMR bacteria to the environment highlight the importance of integrating resident and migratory wild birds in AMR epidemiology to better understand and manage this global public health concern.

Al-Asfar Lake (Yellow Lake) is one of the important shallow wetland lakes in a desert environment in Saudi Arabia that attracted the first inhabitants of this region to settle around the lake waters. The lake is located close to Al-Ahsa Oasis, which is considered the largest and oldest agricultural center in the eastern region of Saudi Arabia. Al-Asfar Lake is a large artificial water body formed from the agriculture and livestock drainage water of the earthen drainage network [17]. The nature of the lake formation is a rear landing area in the huge Arabian Peninsula desert for migratory wild birds [18,19], necessitated the importance of studying the prevalence of AMR in migratory wild birds around the lake to broaden our understanding of antimicrobials dissemination under such an environmental condition.

Although the role of migratory wild birds in the emergence of resistant bacteria is widely recognized in different localities worldwide [3,20,21,22,23], few studies have investigated the role of migratory wild birds in facilitating the transfer of resistant bacteria in Saudi Arabia [24,25,26]. Thus, the present study’s main objective was to determine the presence of antimicrobial-resistant bacteria in samples collected from migratory wild birds around Al-Asfar Lake, with a particular focus on *E. coli* virulence and resistance genes.

## 2. Results

### 2.1. Bacterial Isolates

A total of 132 bacterial isolates were recovered from 113 out of 210 captured migratory wild birds, including 90 (68.2%) *E. coli*, 5 (3.8%) *Salmonella typhimurium* (*S. typhimurium*), and 37 (28%) *Staphylococcus* spp. isolates. *E. coli* and *S. typhimurium* were detected in 90 (42.9%) and 5 (2.4%) birds, respectively, whereas, *Staphylococcus* spp. were detected in 37 (17.6%) birds (Table 1). Mixed infection of *E. coli* and *S. typhimurium* was detected in one (0.5%) bird, while *E. coli* and *Staphylococcus* spp. were detected in 18 (8.6%) birds. The frequency of *E. coli*, *S. typhimurium*, and *Staphylococcus* spp. isolation from each species of the captured wild birds was presented in Table 1.

### 2.2. Antimicrobial Susceptibility Test 

Antimicrobial susceptibility profiles of the 90 *E. coli* and 5 *S. typhimurium* isolates are illustrated in Figure 1a. Whereas, Figure 1b shows the antimicrobial susceptibility profiles of *Staphylococcus* spp. The antimicrobial susceptibility test showed that 41.1% of *E. coli* isolates were resistant to AMP, and 24.4% were resistant to SXT. However, no isolate was resistant to IPM (Table 2). *S. typhimurium* isolates showed resistance to AMP, KAN, DOX, SXT, and CHL (20%, each), and none of the isolates showed resistance to AMC, CTX, IPM, GEN, and CIP (Table 2). *Staphylococcus* isolates showed high frequencies of resistance to PEN (54.1%) and DOX (21.6%), while the lowest number of *Staphylococcus* resistant isolates were observed for CLI (5.4%) and ERY (8.1%) (Table 3).

All *Staphylococcus* isolates were susceptible to AMC, OXA, FOX, and VAN. Figure 2 shows the frequency AMR of *E. coli*, *S. typhimurium*, and *Staphylococcus* spp. isolates in different species of migratory wild birds. *E. coli* and *Staphylococcus* isolates resistance to AMP and PEN, respectively, were the most prevalent type of resistance among the different species of migratory wild birds.

However, no significant difference (*p* > 0.05) was detected between resistance rates of *E. coli* isolated from different species of migratory wild birds. Many significant pairwise correlations were detected between minimum inhibitory concentration values for different antimicrobials against *E. coli* isolated from different migratory wild bird species (Figure 3). The strongest significant (*p* < 0.001) correlation coefficients were detected between AMP and AMC (*r* = 0.62; Ruddy shelduck), Amp and CHL (*r* = 0.77; Common pochard), and AMP and AMC (*r* = 0.86; Little grebe).

Overall, MDR was observed in 13 (14.4%) *E. coli* and 7 (18.9%) *Staphylococcus* spp. isolates, with resistance up to four different antibiotic classes. None of the *S. typhimurium* isolates were MDR. The mean multiple antibiotic resistance (MAR) index was 0.24 (ranged from 0.1 to 0.5) for *E. coli*, 0.1 for *S. typhimurium*, and 0.20 (ranged from 0.09 to 0.45) for *Staphylococcus* spp. Most *E. coli* (78.6%) and *Staphylococcus* spp. (69.6%) isolates showed a MAR index of >0.2. However, all *S. typhimurium* (100%) showed a MAR index of <0.2. The variation between MAR index of *E. coli*, *S. typhimurium*, and *Staphylococcus* spp. recovered from different species of migratory wild birds is presented in Figure 4.

### 2.3. Virulence and Antimicrobial Resistance Genes 

Of the 90 *E. coli* isolates, only 9 (10%) and 5 (5.6%) isolates showed the presence of *eaeA* and *stx2* virulence-associated genes, respectively. However, both *eaeA* and *stx2* genes were identified in four (4.4%) isolates. None of the *E. coli* isolates carried the *hlyA* and *stx1* virulence-associated genes. The frequency of virulence genes in *E. coli* isolates recovered from each migratory wild bird species is presented in Figure 5.

Figure 6 shows the frequency of antimicrobial resistance genes of *E. coli* isolates recovered from each migratory wild bird species. The antimicrobial-resistance gene *bla_CTX-M_* was identified in seven (7.8%) isolates (three isolates were *bla_CTX-M1_* positive, and four isolates were *bla_CTX-M-15_* positive); five (5.6%) isolates expressed *bla_TEM_*, and one (1.1%) isolate expressed *bla_SHV_*. However, both *bla_CTX-M_* and *bla_TEM_* were identified in five (5.6%) isolates, and one (1.1%) isolate carried *bla_CTX-M_* and *bla_SHV_*. Aminoglycosides resistance gene *aac(3)-IV* was detected in eight (8.9%) isolates, whereas the *aadA1* gene was not detected in any isolates. The quinolones (*qnrA*) and tetracycline (*tet(A)*) resistance genes were detected in four (4.4%) and six (6.7%) isolates, respectively. The *mecA* gene was not detected in any of the *Staphylococcus* spp. isolates.

## 3. Discussion

Although migratory wild birds are not implicated directly in the development of antimicrobial resistance since it is not treated with antimicrobial agents, migratory wild birds may act as a reservoir, mixing pot and spreaders of AMR and important indicator for mirroring the impact of human activities (i.e., improper use of antimicrobials) on the environment [11,27,28]. The Arabian Peninsula Desert represents a transit point, especially from August to October and March to May, for birds that migrate all over the distance between Asia, Africa, and Europe, in addition to native wild birds. In deserts, the wetland around oases represents the main landing area for migratory birds where it becomes in close contact with human activities.

In the present study, *E. coli* and *S. typhimurium* were recovered from 42.9% and 2.4% of the collected samples, respectively. The reported *E. coli* positive birds were relatively lower than previously reported in Switzerland (53.7%) [28] and Saudi Arabia (93.0%) [29] and higher than that reported in Singapore (27.1%) [30]. Whereas the prevalence of *Salmonella* positive birds in this study was higher than the 0.99% reported in Singapore [31] and lower than the 12.3% reported in Spain [32]. Only *S. typhimurium* serovar was isolated in the present study that has also been described in wild birds related to animal husbandry as a primary source of infection [32]. The low prevalence of *Salmonella* spp. reported in the current study might be attributed to the collection of samples from apparently healthy migratory wild birds compared to other studies performed on specimens from dead or dying birds [33,34].

*E. coli* frequently used as an indicator for the microbiological quality of water [11,35]. Data in the present study showed a higher incidence of *E. coli* recovery from waterfowl (Common pochard, Little grebe and Ruddy shelduck) comparing to Pied avocet that is relatively less dependent on water. The same finding extends to *S. typhimurium* and *Staphylococcus* recovery. In the same context, 21%, 42%, and 37% of mixed infection cases were recovered from Ruddy shelduck, Common pochard, and little grebe, respectively, whereas no mixed infections were detected in Pied avocet. Previous studies in Saudi Arabia mainly addressed *E. coli* AMR in resident wild birds [29,36].

To better understand AMR prevalence in wildlife, it is important to include multiple bacteria, pathogenic and commensal, with different resistance patterns [30]. In the present study, the phenotypic AMR has been addressed in three major bacterial species; the results revealed 14.4 and 18.9% of the *E. coli* and *Staphylococcus* spp. strains presented MDR. Whereas none of the *Salmonella* strains presented MDR. *Salmonella* is known to be of lower ability to acquire resistance, making it less susceptible to antimicrobial selection pressure than other tested bacteria [30,31]. According to the WHO classification, tetracyclines, penicillins, and sulfonamides were classified as highly important, aminoglycoside was critically important, and cephalosporins (3rd, 4th, and 5th generations) was the highest priority critically important antimicrobials for human medicine [37]. Phenotypically, about (41 and 54%), (26% and 19%), (21%, and 16%), and (14%, and 22%) of *E. coli* and *Staphylococcus* spp. isolates were resistant to penicillins, sulfonamides, aminoglycoside, and tetracycline antibiotics, respectively. In addition, seven *E. coli* isolates (7.8%) were the ESBL-producer based on the phenotypic profile and detection of the *bla_CTX-M_* gene. These results were similar to previous studies where ESBL-producing *E. coli* was first detected in wild birds in Portugal [38], and extended-spectrum cephalosporin-producing Enterobacteriaceae have been isolated from a wide range of bird species across the world [3,27]. None of the isolates recovered in the present study were resistant to carbapenem. While carbapenem resistance is still uncommon in wild animals, there are serious concerns about the emergence of NDM-1 and IMP carbapenemases in wild birds [21,39]. This goes in context with the antibiotic resistance pattern of bacteria isolated from water spring, which is the origin of the Al-Asfar Lake where 76.9%, 65.4%, and 50% of bacterial isolates are resistant to penicillins, aminoglycoside, and tetracycline, respectively [40]. This may explain why birds (i.e., Pied Avocet) with relatively lower water dependence showed significantly lower resistance to all antibiotics.

It is worth noting that the highest prevalence of antibiotic-resistant bacteria has been recorded in aquatic-associated birds, which agrees with previous findings [3,27]. Thus, the significantly higher antibiotic resistance in *E. coli* isolates recovered from Ruddy shelduck in the present study may be related to their feeding and living habits. However, fecal samples collected from Ruddy shelduck around Qinghai Lake, China, showed weak antibiotic resistance to *E. coli* [16]. There are many reasons for differences in antimicrobial resistance in the normal microbiota of migratory wild birds. First, resistance can evolve de novo through spontaneous mutation (s) [41]. Second, horizontal gene transfer from other microbes can develop resistance; certain bacteria and fungi represent natural sources of genes for drug resistance and can function as reservoirs in the environment [42]. Third, bacteria with antimicrobial drug resistance could be introduced into the area either by migratory birds or by human waste (food and excretion) from local fishermen, settlers, and prospectors.

In the present study, the *eaeA* and *stx2* virulence-associated genes were identified in 10%, and 5.6% of the *E. coli* isolates, respectively. This result was consistent with Kobayashi et al. [43], who identified *eaeA* and *stx2* in Japan. However, the incidence of the *eaeA* and *stx2* reported in the present study was higher than the 2.3% reported in wild brides in Iran [44]. Several studies have reported the association of *eaeA* and *stx2* genes, which indicated the importance of testing *eaeA* positive isolates for the presence of the *stx2* gene [44,45]. The *hlyA* and *stx1* genes were not identified in any *E. coli* recovered in the present study. This result contrasts with a recent study carried out in wild birds in Central Italy, which detected 3.3% and 8.3% of birds were positive for *hlyA* and *stx1*, respectively [46].

In the present study, 7.8%, 5.6%, and 1.1% of *E. coli* isolates carried *bla_CTX-M_*, *bla_TEM_*, and *bla_SHV_* genes, respectively. These results agree with several reports indicated the presence of ESBL-producing bacteria among migratory wild birds [47,48,49]. The alarming levels of ESBL-producing *E. coli* recovered from the present study were lower than that reported in wild birds in countries as Spain (74.8%), Netherlands (37.8%), England (27.1%), Sweden (20.7%), Latvia (17.4%) and Portugal (12.7%), and higher than that reported in Portugal (12.7%), Ireland (4.5%), Poland (0.7%), and Denmark (0.0%) [50]. Furthermore, about 8.9%, 4.4%, and 6.7% of the recovered *E. coli* isolates possess plasmid-mediated quinolone resistance (*qnrA*), aminoglycosides resistance genes *aac(3)-IV* that encode acetyltransferases enzyme, and tetracycline resistance gene (*tet(A)*) that is often associated with mobile elements, respectively. Although our study design cannot confirm the source of ESBL-producing isolates that recovered in migratory wild birds, our results represent further evidence for the potential role of migratory wild birds in the global dissemination of ESBL that poses a serious challenge to the globe.

It should be noted that our study has two limitations. First, unequally collected samples from different wild bird species; second, there are no environmental samples collected. However, in a previous study, 86.7% of water samples collected from different Al-Ahsa water springs were positive for *E. coli* [40], indicating the central role of anthropogenic impact in the area where wild birds live, feed, and drink [39,51].

## 4. Materials and Methods

### 4.1. Study Area 

The study was carried out in Al-Asfar Lake, located 13 km east of Al-Ahsa Oasis (N25 33 54, E49 50 15) Saudi Arabia. The lake extends over 2170 ha close to the Arabian Gulf. Al-Asfar Lake is an important bird area that provides shelter for a wide diversity of migratory wild birds, especially during the winter season. Al-Asfar Lake includes an alpine vegetation area with winding boundaries of watered areas followed by sandy surroundings. Varied bird species have been observed in the wetland, including large birds like ducks and geese to sparrows and small birds.

### 4.2. Birds and Sampling 

Birds were captured within a 500 m radius vegetated area. The captured birds were described and named, according to Porter and Aspinall [52]. A pair of cloacal swabs were collected from each bird (a sterile swab was inserted into the captured bird’s cloaca and then rotated to take the fluid sample). The first swab was used for screening the avian influenza virus by a rapid test (FluDETECT™ Avian, Zoetis, Kalamazoo, MI, USA), and the second swab was stored in a sterile tube containing 5 mL of buffered peptone water (BPW; Oxoid, UK) for later bacteriological examination.

A total of 210 birds of different species, including Ruddy shelduck (*Tadorna ferruginea*; *n* = 70), Common pochard (*Aythya ferina*; *n* = 50), Pied avocet (*Recurvirostra avosetta*; *n* = 30), and Little grebe (*Tachybaptus ruficollis*; *n* = 60) were captured and sampled between January and December 2016. After sampling, all birds were allowed to fly to their natural habitat freely. All collected swabs were tested negative for avian influenza antigen by the rapid test and then transported in an icebox at 4 °C to the laboratory for bacteriological examination.

### 4.3. Bacterial Isolation and Identification

Tubes containing swabs and BPW were gently mixed. For isolation of *E. coli* and *Staphylococcus* spp., 100 μL from each tube was streaked onto each of MacConkey, Sorbitol MacConkey, and Baird-Parker (Oxoid, UK), then incubated at 37 °C for 24 h. Lactose fermenting colonies on MacConkey agar, white colonies on Sorbitol MacConkey agar, and black colonies on Baird–Parker agar were identified biochemically to species level by VITEK^®^ 2 COMPACT (BioMérieux, France). For isolation of *Salmonella*, tubes containing BPW were enriched overnight aerobically at 37 °C, then incubated on Rappaport-Vassiliadis broth (Oxoid, UK) at 42 °C in aerobic conditions for 24 h, before inoculation on to xylose lysine deoxycholate agar (Oxoid, UK) and incubated under aerobic conditions at 37 °C for 24 h. Suspected colonies were identified biochemically by VITEK^®^ 2 COMPACT (BioMérieux, France). Biochemically identified *Salmonella* isolates have been serologically confirmed on the basis of somatic (O) and flagellar (H) antigens by slide agglutination using commercial antisera (SISIN, Germany) following the Kauffman–White scheme [53].

For molecular conformation, bacterial DNA was extracted from biochemically identified *E. coli*, *Salmonella*, and *Staphylococcus* isolates for amplification and sequencing of 16S rRNA gene according to Lane [54] and Weisberg et al. [55].

### 4.4. Antimicrobial Susceptibility Test

The standard disk diffusion test on Mueller Hinton agar (MHA) using cefotaxime (30 µg) and cefoxitin (30 µg) disk was performed according to the guidelines of the Clinical Laboratory Standards Institute (CLSI) [56] to identify extended-spectrum β-lactamase and methicillin-resistance bacteria, respectively. Two different sets of antimicrobials were selected for *E. coli*/*Salmonella* and *Staphylococcus* antimicrobials susceptibility testing. Antimicrobials include ampicillin (AMP), penicillin (PEN), amoxicillin-clavulanate (AMC), cefotaxime (CTX), oxacillin (OXA), imipenem (IPM), cefoxitin (FOX), kanamycin (KAN), gentamicin (GEN), doxycycline (DOX), ciprofloxacin (CIP), trimethoprim-sulfamethoxazole (SXT), chloramphenicol (CHL), erythromycin (ERY), clindamycin (CLI), and vancomycin (VAN). The minimum inhibitory concentration (MIC) was determined by double fold dilution of antimicrobials (0.125–256 μg/mL) as recommended by the CLSI [56].

The dilutions and breakpoints were defined according to the CLSI [56]. Isolates were classified as resistant (R) or intermediate (I), or susceptible (S) based on the MIC breakpoint values. Multidrug-resistant (MDR) was considered when isolates were resistant to three or more different antimicrobial classes [57]. Furthermore, the MAR index was determined for all isolates according to the protocol described by Krumperma, [58] using the formula a/b (where “a” refers to the number of antimicrobials to which the isolate was resistant, and “b” represents the total number of antimicrobials to which the isolate was exposed).

### 4.5. Virulence and AMR Genes 

The extracted DNA of *E. coli* isolates was amplified for identification of intimin (encoded by *eaeA* gene), enterohemolysin (encoded by *hlyA* gene), and Shiga toxins (encoded by *stx1* and *stx2* genes) [59,60]. AMR genes investigated were *bla_CTX-M_*, *bla_TEM_*, and *bla_SHV_* for extended-spectrum-β-lactamase (ESBL) [61,62]; *aac(3)-IV* and *aadA1* for aminoglycosides resistance [63]; *qnrA* for quinolones resistance [64]; and *tet(A)* for tetracycline resistance [65].

For *Staphylococcus* isolates, the AMR gene investigated was the *mecA* gene for methicillin resistance [66]. In each PCR assay, positive and negative controls were used. The primers, the annealing temperature, and the expected size of the DNA product for each of the investigated genes are shown in Table 4.

### 4.6. Data Analysis

Data were visualized with R software (R Core Team, 2019; version 3.5.3). A heatmap based on the isolate’s antimicrobial resistance profiles was built using the “Complex-Heatmap” R package [67]. Fisher’s exact test was used to identify the difference in *E. coli* resistance rate between migratory wild bird species. However, the correlation among antimicrobial MIC values for *E. coli* isolates recovered from each migratory wild bird species was assessed using the Spearman’s rank correlation test.

## 5. Conclusions

The higher bacterial recovery, antimicrobial resistance (phenotype, genotype, and MDR) from all samples collected from migratory wild birds around Al-Asfar Lake indicated environmental dissemination of antimicrobial resistance to wild birds that can maintain and spread the resistance bacteria along their migration route. That mirroring human activity and its impact on the environment. Migratory wild bird feeding and inhabitant habits are the main driving force of antimicrobial resistance in different wild bird species, explaining why certain migratory wild bird species can acquire, share in selection pressure, and disseminate the antibiotic-resistant bacteria. Although all available evidence in the present study ensures the implication of contaminated water in AMR incidence in migratory wild birds, it cannot rule out other sources including the abroad one. This highlights the importance of collecting samples from different environmental samples from and around the lake along with samples from water from irrigation canals and treated wastewater from nearby sewage stations to conduct a co-occurring network analysis.

## Figures and Tables

**Figure 1 antibiotics-10-00260-f001:**
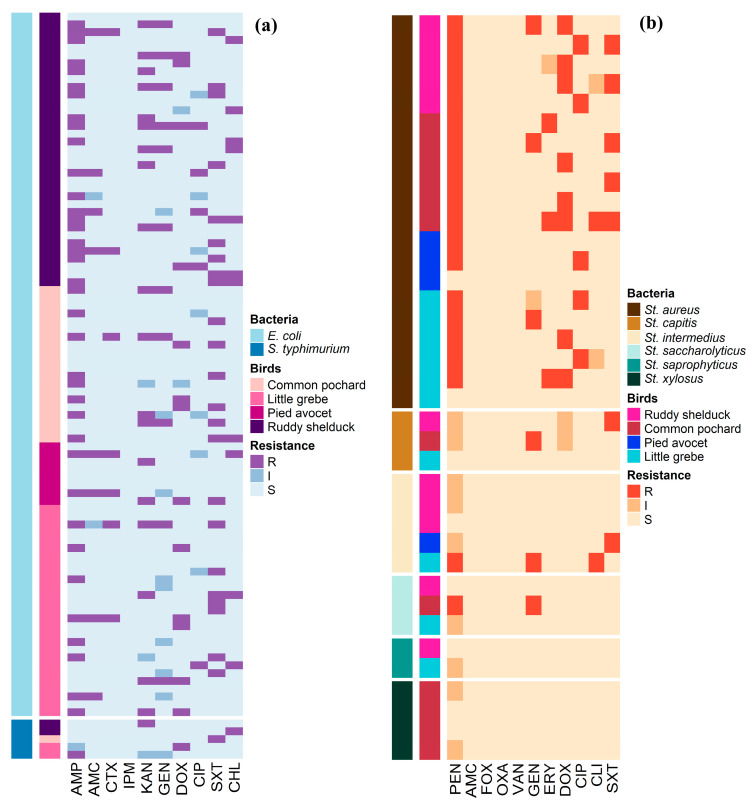
Distribution and clustering of *Escherichia coli*, *Salmonella typhimurium*, and *Staphylococcus* isolates recovered from different species of migratory wild birds around the Al-Asfar Lake. (**a**) Heat map representation of antimicrobial-resistant profiles of the 90 *Escherichia coli* and 5 *Salmonella typhimurium* isolates. (**b**) Heat map representation of antimicrobial-resistant profiles of the 37 *Staphylococcus* isolates.

**Figure 2 antibiotics-10-00260-f002:**
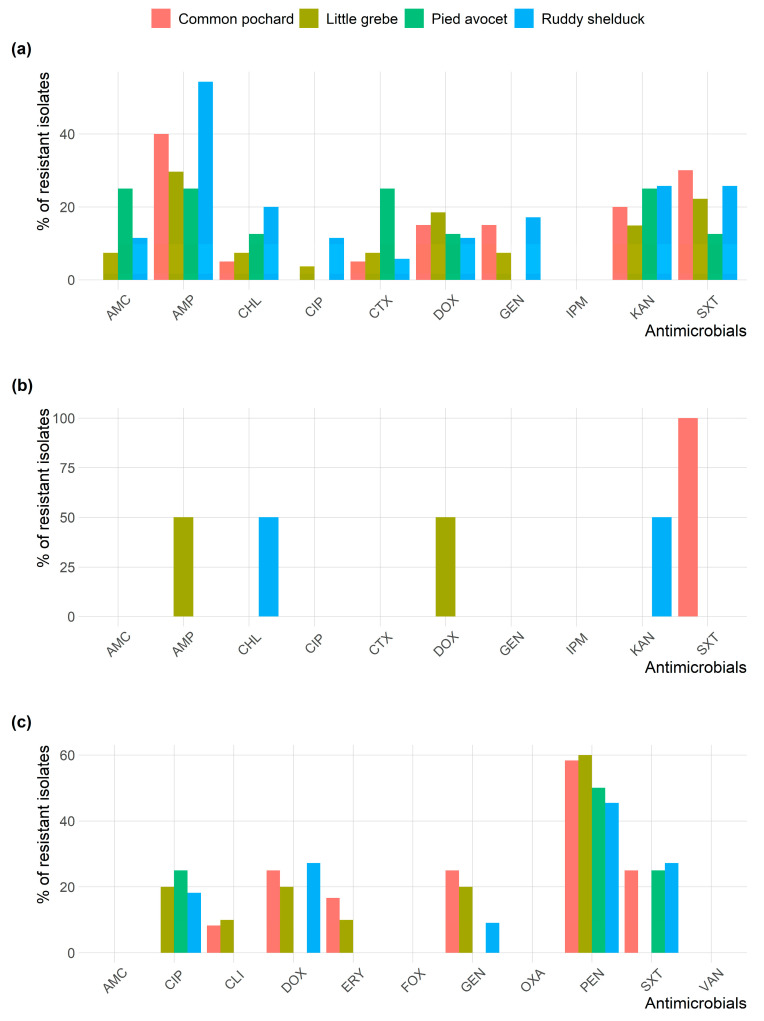
Frequency of antimicrobial resistance of (**a**) *Escherichia coli*; (**b**) *Salmonella typhimurium*; and (**c**) *Staphylococcus* spp. recovered from different species of migratory wild birds around the Al-Asfar Lake.

**Figure 3 antibiotics-10-00260-f003:**
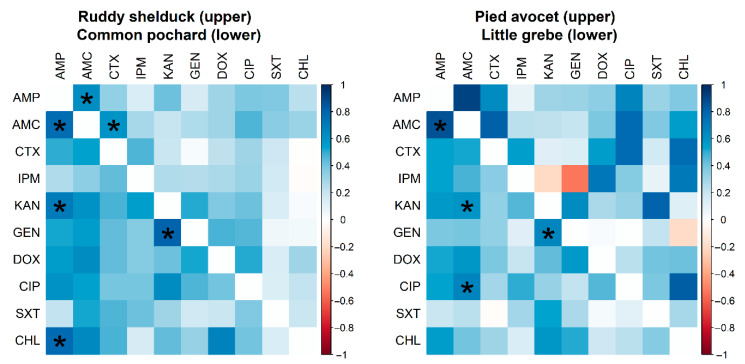
Spearman rank correlation test results based on the minimum inhibitory concentrations of *Escherichia coli* (*n* = 90) isolates recovered from different species of migratory wild birds for ten antimicrobials. The blue color indicated a positive correlation, and the red shows a negative correlation. Strikes (*****) indicate significance at *p* < 0.001.

**Figure 4 antibiotics-10-00260-f004:**
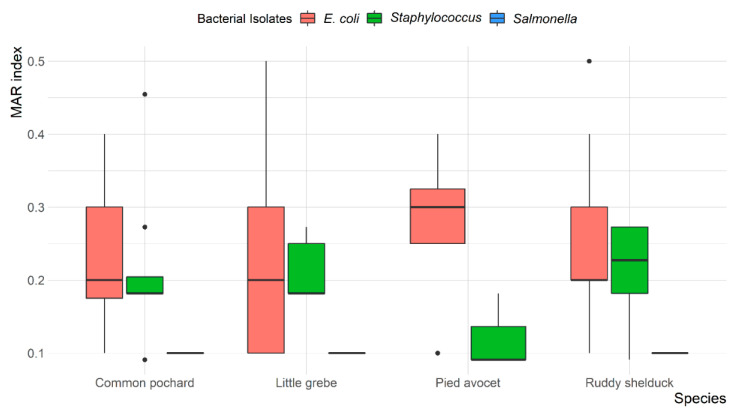
Box and whisker plot of multiple antibiotic resistance (MAR) index among *Escherichia coli*, *Salmonella typhimurium*, and *Staphylococcus* spp. recovered from different species of migratory wild birds around the Al-Asfar Lake.

**Figure 5 antibiotics-10-00260-f005:**
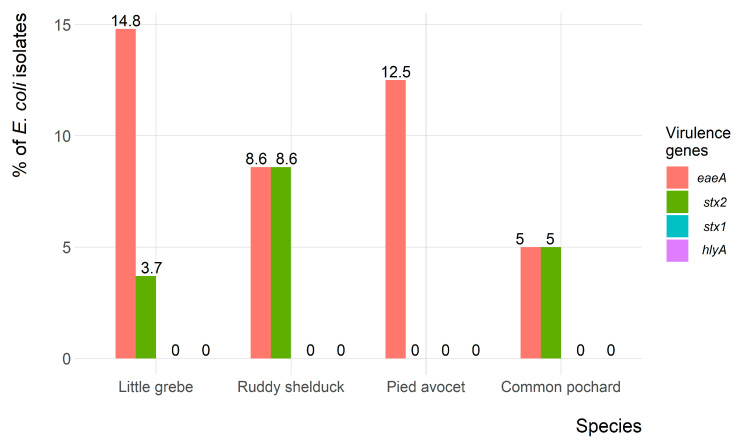
Frequency of virulence genes of *Escherichia coli* (*n* = 90) isolates recovered from migratory wild birds around the Al-Asfar Lake.

**Figure 6 antibiotics-10-00260-f006:**
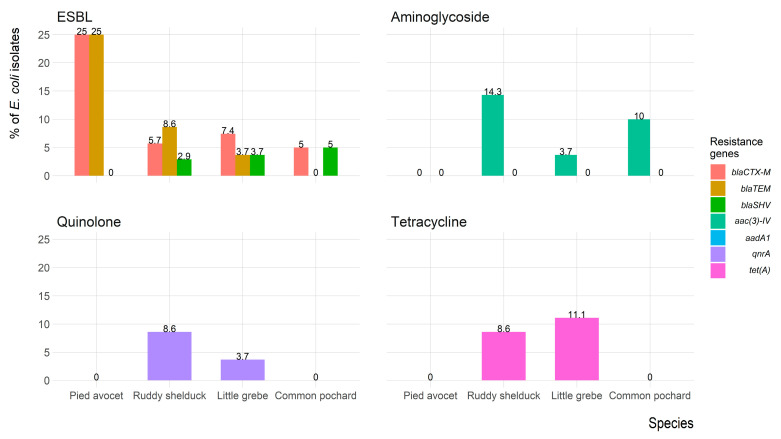
Frequency of antimicrobial resistance genes of *Escherichia coli* (*n* = 90) isolates recovered from migratory wild birds around the Al-Asfar Lake.

**Table 1 antibiotics-10-00260-t001:** Number and percentage of *Escherichia coli*, *Salmonella typhimurium*, and *Staphylococcus* spp. isolates recovered from different species of migratory wild birds around Al-Asfar Lake.

Bacteria Species	No. (%) of Bacterial Isolated				Total (*n* = 210)
	Common Pochard (*n* = 50)	Pied Avocet (*n* = 30)	Little Grebe (*n* = 60)	Ruddy Shelduck (*n* = 70)	
*E. coli*	20 (40.0)	8 (30.0)	27 (45.0)	35 (50.0)	90 (42.9)
*Salmonella*					
*S. typhimurium*	1 (2.0)	0 (0.0)	2 (3.3)	2 (2.9)	5 (2.4)
*Staphylococcus*					37 (17.6)
*St. aureus*	6 (12.0)	3 (10.0)	6 (10.0)	5 (7.1)	20 (9.5)
*St. intermedius*	0 (0.0)	1 (3.3)	1 (1.7)	3 (4.3)	5 (2.4)
*St. xylosus*	4 (8.0)	0 (0.0)	0 (0.0)	0 (0.0)	4 (1.9)
*St. capitis*	1 (2.0)	0 (0.0)	1 (1.7)	1 (1.4)	3 (1.4)
*St. saccharolyticus*	1 (2.0)	0 (0.0)	1 (1.7)	1 (1.4)	3 (1.4)
*St. saprophyticus*	0 (0.0)	0 (0.0)	1 (1.7)	1 (1.4)	2 (1.0)

**Table 2 antibiotics-10-00260-t002:** The antimicrobial-resistant profiles of *Escherichia coli* (*n* = 90) and *Salmonella typhimurium* (*n* = 5) isolates recovered from migratory wild birds around the Al-Asfar Lake.

Antimicrobials			No. of Resistant *E. coli* Isolates (%)					No. of Resistant *Salmonella* Isolates (%)				
Rank ^1^	Class	Agents	*n*	Common Pochard	Pied Avocet	Little Grebe	Ruddy Shelduck	*n*	Common Pochard	Pied Avocet	Little Grebe	Ruddy Shelduck
II	Penicillins	PEN	37	8 (21.6)	2 (5.4)	8 (21.6)	19 (51.4)	1	−	−	1 (100.0)	−
		AMC	8	0 (0.0)	2 (25.0)	2 (25.0)	4 (50.0)	0	−	−	−	−
I	Cephalosporins	CTX	7	1 (14.3)	2 (28.6)	2 (28.6)	2 (28.6)	0	−	−	−	−
I	Carbapenem	IPM	0	0 (0.0)	0 (0.0)	0 (0.0)	0 (0.0)		−	−	−	−
I	Aminoglycoside	KAN	19	4 (21.1)	2 (10.5)	4 (21.1)	9 (47.4)	1	−	−	−	1 (100.0)
		GEN	11	3 (27.3)	0 (0.0)	2 (18.2)	6 (54.5)	0	−	−	−	−
II	Tetracycline	DOX	13	3 (23.1)	1 (7.7)	5 (38.5)	4 (30.8)	1	−	−	1 (100.0)	−
I	Quinolones	CIP	5	0 (0.0)	0 (0.0)	1 (20.0)	4 (80.0)	0	−	−	−	−
II	Sulfonamide	SXT	23	6 (26.1)	1 (4.3)	7 (30.4)	9 (39.1)	1	−	−	−	−
II	Amphenicols	CHL	11	1 (9.1)	1 (9.1)	2 (18.2)	7 (63.6)	1	−	−	−	1 (100.0)

^1^ Rank I, critically important; rank II, highly important (based on World Health Organization’s categorization).

**Table 3 antibiotics-10-00260-t003:** The antimicrobial-resistant profile of *Staphylococcus* (*n* = 37) isolates recovered from migratory wild birds around the Al-Asfar Lake.

Antimicrobials			No. of Resistant *Staphylococcus* Isolates (%)				
Rank ^1^	Class	Agents	*n*	Common Pochard	Pied Avocet	Little Grebe	Ruddy Shelduck
II	Penicillins	PEN	20	7 (35.0)	2 (10.0)	6 (30.0)	5 (25.0)
		AMC	−	−	−	−	−
		OXA	−	−	−	−	−
I	Cephalosporins	FOX	−	−	−	−	−
I	Glycopeptides	VAN	−	−	−	−	−
I	Aminoglycoside	GEN	6	3 (50.0)	−	2 (33.3)	1 (16.7)
I	Macrolide	ERY	3	2 (66.7)	−	1 (33.3)	−
II	Tetracycline	DOX	8	3 (37.5)	−	2 (25.0)	3 (37.5)
I	Quinolones	CIP	5	−	1 (20.0)	2 (40.0)	2 (40.0)
II	Lincosamides	CLI	2	1 (50.0)	−	1 (50.0)	−
II	Sulfonamide	SXT	7	3 (42.9)	1 (14.3)	−	3 (42.9)

^1^ Rank I, critically important; rank II, highly important (based on World Health Organization’s categorization).

**Table 4 antibiotics-10-00260-t004:** Primers, product size, and annealing temperatures used for virulence and antimicrobial resistance genes identification in the present study.

Gene	Primer Sequences	Product Size (bp)	Annealing (°C)	Ref.
*stx1*	fw: 5′-AAATCGCCATTCGTTGACTACTTCT-3′	370	60	[59]
	rev: 5′-TGCCATTCTGGCAACTCGCGATGCA-3′			
*stx2*	fw: 5′-CAGTCGTCACTCACTGGTTTCATCA-3′	283	60	[59]
	rev: 5′-GGATATTCTCCCCACTCTGACACC-3′			
*hlyA*	fw: 5′-GGTGCAGCAGAAAAAGTTGTAG-3′	1551	57	[60]
	rev: 5′-TCTCGCCTGATAGTGTTTGGTA-3′			
*eaeA*	fw: 5′-CCCGAATTCGGCACAAGCATAAGC-3′	863	52	[60]
	rev: 5′-TCTCGCCTGATAGTGTTTGGTA-3′			
*bla_CTX-M-I_*	fw: 5′-GACGATGTCACTGGCTGAGC-3′	499	55	[62]
	rev: 5′-AGCCGCCGACGCTAATACA- 3′			
*bla_CTX-M-II_*	fw: 5′-GCGACCTGGTTAACTACAATCC-3′	351	55	[62]
	rev: 5′-CGGTAGTATTGCCCTTAAGCC -3′			
*bla_CTX-M-III_*	fw: 5′-CGCTTTGCCATGTGCAGCACC -3′	307	55	[62]
	rev: 5′-GCTCAGTACGATCGAGCC -3′			
*bla_CTX-M-IV_*	fw: 5′-GCTGGAGAAAAGCAGCGGAG-3′	474	62	[62]
	rev: 5′-GTAAGCTGACGCAACGTCTG -3′			
*bla_TEM_*	fw: 5′-GAGTATTCAACATTTTCGT -3′	857	58	[63]
	rev: 5′-ACCAATGCTTAATCAGTGA -3′			
*bla_SHV_*	fw: 5′-TCGCCTGTGTATTATCTCCC-3′	768	52	[63]
	rev: 5′-CGCAGATAAATCACCACAATG-3′			
*aac(3)-IV*	fw: 5′-CTTCAGGATGGCAAGTTGGT-3′	286	55	[63]
	rev: 5′-TCATCTCGTTCTCCGCTCAT-3′			
*aadA1*	fw: 5′-TATCCAGCTAAGCGCGAACT-3′	447	58	[63]
	rev: 5′-ATTTGCCGACTACCTTGGTC-3′			
*qnrA*	fw: 5′-GGGTATGGATATTATTGATAAAG-3′	670	50	[64]
	rev: 5′-CTAATCCGGCAGCACTATTTA-3′			
*tet(A)*	fw: 5′-GGTTCACTCGAACGACGTCA-3′	577	57	[65]
	rev: 5′-CTGTCCGACAAGTTGCATGA-3′			
*mecA*	fw: 5′-AAAATCGATGGTAAAGGTTGGC-3′	530	55	[66]
	rev: 5′-AG TTCTGCAGTACCGGATTTGC-3′			

## Data Availability

The data presented in this study are available on request from the corresponding author.

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
