# Peer review of "Migratory Wild Birds as a Potential Disseminator of Antimicrobial-Resistant Bacteria around Al-Asfar Lake, Eastern Saudi Arabia"

_antibiotics, 2021, doi:10.3390/antibiotics10030260_

Round 1

Reviewer 1 Report

In this work, the authors studied the antimicrobial-resistant bacteria in samples collected from migratory wild birds around Al-Asfar Lake. The originality, experimental work and scientific background sound good to consider for acceptance for publication. However, there are a couple of issues that should be addressed.
1. The abstract should state briefly the purpose of the research, the principal results and major conclusions. Please refine the current abstract.
2. It is unclear why this work presents a particular focus on E. coli virulence and resistance genes?
3. I am wondering how to ensure the captured birds were migratory rather than the non-migratory wild birds? More information should be provided because both migratory and non-migratory wild birds can act as reservoirs of bacteria, such as E. coli.
4. No statistical analysis was performed in this work.

Author Response

Reviewer 1:

---------------

R1.1. In this work, the authors studied the antimicrobial-resistant bacteria in samples collected from migratory wild birds around Al-Asfar Lake. The originality, experimental work and scientific background sound good to consider for acceptance for publication. However, there are a couple of issues that should be addressed.

AU: Authors thank the reviewer for considering our manuscript for publication. We have incorporated the following reviewer’s comments in the revised version of the manuscript.   

R1.2. The abstract should state briefly the purpose of the research, the principal results and major conclusions. Please refine the current abstract.

AU: The abstract was edited, and the study's main objective was added as suggested (Lines 34-36).

R1.3. It is unclear why this work presents a particular focus on E. coli virulence and resistance genes?

AU: Authors thank the reviewer for raising this. The study gives particular focus on E. coli virulence and resistance genes for the following reasons:

  • coli isolates were the most common bacteria recovered from birds involved in the study.
  • The β-lactamase resistance E. coli is of higher interest as various studies have reported the association between β-lactamase resistant E. coli isolated from wild bird/animals and humans in Saudi Arabia.

R1.4. I am wondering how to ensure the captured birds were migratory rather than the non-migratory wild birds? More information should be provided because both migratory and non-migratory wild birds can act as reservoirs of bacteria, such as E. coli.

AU: Authors agree with the reviewer’s comment that resident and migratory wild birds can act as reservoirs and disseminators of AMR bacteria. However, the species involved in this study mainly migratory and are not from the resident wild birds in Saudi Arabia based on the Ministry of Environment, Water and Agriculture in Saudi Arabia. Also, these birds were listed on the different migratory routes over Asia, Africa, and Europe.

R1.5. No statistical analysis was performed in this work.

AU: The reviewer raises a good point. The authors agree with the reviewer's comment that statistical analysis and correlation among antimicrobials would add to our study's quality. We identify the differences in the resistance rate among E. coli isolates recovered from different migratory wild bird species. Also, the strength of correlation among antimicrobial MIC values in each species (lines 139 – 149 and lines 355 – 358).  

Reviewer 2 Report

I have enjoyed reading your article on the potential role of  migratory wild birds to disseminate antimicrobial-resistant bacteria around Al-Asfar Lake, Eastern Saudi Arabia.

Please find my suggestions bellow.

General comments:

1). Please define what type of E. coli did you isolate. Pathogenic E. coli in birds are called avian pathogenic E. coli (APEC), but assuming that you isolated your E. coli from healthy birds, these are mostly commensal (generic) E. coli.

 Somehow it is surprising that the isolation rate was only 42.9%, because generic E. coli usually can be recovered from almost all of the birds.

2) It would add to the level of the analysis to add statistical tests when you compare AMR among various migratory bird types. Also, correlation among antimicrobials and the description of multidrug resistance patterns would add to the quality of your study.

Please read the following articles, where the authors used some of the previously mentioned methods:

Varga C, Guerin MT, Brash ML, Slavic D, Boerlin P, Susta L. Antimicrobial resistance in fecal Escherichia coli and Salmonella enterica isolates: a two-year prospective study of small poultry flocks in Ontario, Canada. BMC Vet Res. 2019 Dec 21;15(1):464. doi: 10.1186/s12917-019-2187-z. 

 Varga C, Brash ML, Slavic D, Boerlin P, Ouckama R, Weis A, Petrik M, Philippe C, Barham M, Guerin MT. Evaluating Virulence-Associated Genes and Antimicrobial Resistance of Avian Pathogenic Escherichia coli Isolates from Broiler and Broiler Breeder Chickens in Ontario, Canada. Avian Dis. 2018 Sep;62(3):291-299. doi: 10.1637/11834-032818-Reg.1.

Specific comments:

1) "Line 64 - AMR is a dynamic and multifaceted human, animal, and environmental problem" I think you should include "health problem". You should mention also the "One Health" concept.

2) Line 85 – MAR should be changed to  AMR

3) Line 137 - "mean MAR index" should be changed to AMR index

4) Line 171 & Line 188 " mirroring the impact of human activities on the environment" Please be more specific. What type of human activities are you talking about? I assume that you are referring to the improper antimicrobial use, contamination of groundwater and lakes with antimicrobial residues etc.

Thank you for considering my suggestions.

Author Response

Reviewer 2:

---------------

R2.1. I have enjoyed reading your article on the potential role of  migratory wild birds to disseminate antimicrobial-resistant bacteria around Al-Asfar Lake, Eastern Saudi Arabia. Please find my suggestions bellow.

AU: The authors thank the reviewer for considering our manuscript interesting and enjoyable in reading. We have incorporated the following reviewer’s suggestions in the revised version of the manuscript.

General comments:

R2.2. Please define what type of E. coli did you isolate. Pathogenic E. coli in birds are called avian pathogenic E. coli (APEC), but assuming that you isolated your E. coli from healthy birds, these are mostly commensal (generic) E. coli.

AU: Yes, the E. coli isolates were recovered from apparently healthy birds. However, a number of E. coli isolates were defined as pathogenic (Shiga toxin-producing) based on virulence gene detection.

R2.3. Somehow it is surprising that the isolation rate was only 42.9%, because generic E. coli usually can be recovered from almost all of the birds.

AU: We understand the reviewer's concern about the isolation rate (42.9%). However, we would like to confirm that the isolates were recovered from cloacal swabs (not intestinal content). Furthermore, the isolation rate was higher or similar to the previous studies' rate (Zurfluh, et al., 2019; Ong, et al., 2020).

R2.4. It would add to the level of the analysis to add statistical tests when you compare AMR among various migratory bird types. Also, correlation among antimicrobials and the description of multidrug resistance patterns would add to the quality of your study. Please read the following articles, where the authors used some of the previously mentioned methods:

  • Varga C, Guerin MT, Brash ML, Slavic D, Boerlin P, Susta L. Antimicrobial resistance in fecal Escherichia coli and Salmonella enterica isolates: a two-year prospective study of small poultry flocks in Ontario, Canada. BMC Vet Res. 2019 Dec 21;15(1):464. doi: 10.1186/s12917-019-2187-z.
  • Varga C, Brash ML, Slavic D, Boerlin P, Ouckama R, Weis A, Petrik M, Philippe C, Barham M, Guerin MT. Evaluating Virulence-Associated Genes and Antimicrobial Resistance of Avian Pathogenic Escherichia coli Isolates from Broiler and Broiler Breeder Chickens in Ontario, Canada. Avian Dis. 2018 Sep;62(3):291-299. doi: 10.1637/11834-032818-Reg.1.

AU: The reviewer raises a good point. The authors agree with the reviewer's comment that statistical analysis and correlation among antimicrobials would add to our study's quality. We identify the differences in the resistance rate among E. coli isolates recovered from different migratory wild bird species. Also, the strength of correlation among antimicrobial MIC values in each species (lines 139 – 149 and lines 355 – 358).  

Specific comments:

R2.5. "Line 64 - AMR is a dynamic and multifaceted human, animal, and environmental problem" I think you should include "health problem". You should mention also the "One Health" concept.

AU: Changed as suggested (Line 67).

R2.6. Line 85 – MAR should be changed to  AMR

AU: Corrected (Line 88).

R2.7. Line 137 - "mean MAR index" should be changed to AMR index

AU: This one is an abbreviation is correct as (MAR = multiple antibiotic resistance index). The abbreviation's full name has been added to be clear for the reader (Line 140).  

R2.8. Line 171 & Line 188 " mirroring the impact of human activities on the environment" Please be more specific. What type of human activities are you talking about? I assume that you are referring to the improper antimicrobial use, contamination of groundwater and lakes with antimicrobial residues etc.

AU: These precisely what we meant by human activities. An example was added to be clear for the readers (Line 187). 

This manuscript is a resubmission of an earlier submission. The following is a list of the peer review reports and author responses from that submission.